# Canadian historical Snow Water Equivalent dataset (CanSWE, 1928-2020)

Vincent Vionnet[1], Colleen Mortimer[2], Mike Brady [2], Louise Arnal[3], Ross Brown [2]

[1]Meteorological Research Division, Environment and Climate Change Canada, Dorval, Canada
[2]Climate Research Division, Environment and Climate Change Canada, Toronto, Canada
[3]University of Saskatchewan Coldwater Laboratory, Canmore, Alberta, Canada

*Correspondence to*: Vincent Vionnet (vincent.vionnet@ec.gc.ca)

**Abstract.**

In situ measurements of water equivalent of snow cover (SWE) – the vertical depth of water that would be obtained if all the snow cover melted completely– are used in many applications including water management, flood forecasting, climate monitoring, and evaluation of hydrological and land surface models. The Canadian historical SWE dataset (CanSWE) combines manual and automated pan-Canadian SWE observations collected by national, provincial and territorial agencies as well as hydropower companies. Snow depth (SD) and  bulk snow density (defined as the ratio of SWE to SD) are also included when available. This new dataset supersedes the previous Canadian Historical Snow Survey (CHSSD) dataset published by Brown et al. (2019), and this paper describes the efforts made to correct metadata, remove duplicate observations, and quality control records. The CanSWE dataset was compiled from 15 different sources and includes SWE information for all provinces and territories that measure SWE. Data were updated to July 2020 and new historical data from the Government of Northwest Territories, Government of Newfoundland and Labrador, Saskatchewan Water Security Agency, and Hydro Quebec were included. CanSWE includes over one million SWE measurements from 2607 different locations across Canada over the period 1928 – 2020. It is publicly available at  https://doi.org/10.5281/zenodo.4734371 (Vionnet et al., 2021).

## 1 Introduction

Reliable in situ information of snow water equivalent (SWE) or more precisely water equivalent of snow cover according to WMO (2018) – the vertical depth of water that would be obtained if the snow cover melted completely, which equates to the snow-cover mass per unit area (WMO, 2018) – is critical for flood and drought predictions (e.g., Jörg-Hess et al., 2015; Berghuijs et al., 2016; Vionnet et al., 2020), streamflow management of water supply for hydropower generation (e.g., Magnusson et al., 2020), irrigation planning (e.g., Biemans et al., 2019) and is a key environmental variable for climate monitoring and understanding (e.g., Clark et al., 2001; Brown et al., 2019). In situ SWE measurements can be made manually or via automatic sensors (Kinar and Pomeroy, 2015). Manual SWE measurements typically consist of single point

measurement (snow pit or single measurement carried out with a snow tube) or multi-point gravimetric snow surveys (also known as snow transects or snow courses) collected along a pre-determined transect (WMO, 2018, Lopez Moreno et al., 2020). Manual snow surveys are generally representative of the prevailing land cover and terrain but are time-consuming and expensive which limits their temporal frequency, especially in remote locations. Automatic stations can overcome this limitation and provide SWE measurements at a higher temporal frequency but have the disadvantage of only measuring SWE at a single point. Snow pillows (Beaumont, 1965) and snow scales (Johnson, 2004, Smith et al, 2017) automatically measure SWE from the overlying pressure and weight of the snowpack, respectively. Indirect methods using passive radiation sensors installed below or above the snowpack have also been developed. They measure the attenuation by the snowpack of natural cosmic radiation (Kodama et al., 1979; Paquet et al., 2008) or naturally emitted gamma radiation from the soil (Choquette et al., 2013). Finally, SWE can be automatically derived by analysis of the signal from Global Navigation Satellite System receivers (Henkel et al., 2018, Steiner et al., 2019).

SWE observation networks using different measurements methods have been deployed at a national scale in various countries to provide valuable in situ information. Russia maintains a vast long-term network of manual snow survey transects located in the vicinity of meteorological stations (Bulygina et al., 2011). National SWE measurements relying on manual methods are also available in several European countries: Finland, Estonia, Ukraine and Turkey use for example snow courses whereas countries such as Germany or Czech Republic rely on single-point measurements (Haberkorn, 2019). In the Western United States (US), manual SWE measurements are collected along permanent snow courses maintained by the US Department of Agriculture (Natural Resources Conservation Service, 1988) and in the Northeast by various state departments (McKay et al., 1994). Another source of SWE information in the Western US and Alaska is the snowpack telemetry (SNOTEL) network using automatic snow pillows (Serreze et al., 1999). In situ SWE data from several of these networks are used for a number of research and development applications. For example, they serve as reference data for the evaluation of a variety of large-scale gridded SWE products (e.g., Mortimer et al., 2020) including (i) snowpack models driven by meteorological reanalysis (e.g., Brun et al., 2013) (ii) passive microwave estimates combined with surface snow depth observation such as the GlobSnow product (Pulliainen et al., 2020) and (iii) regional climate models (e.g., Rasmussen et al., 2011). Gridded snow products can also be derived from manual and automatic in situ SWE measurements (e.g., Brown et al., 2019). In a hydrological context, SWE measurements from large-scale networks can inform the calibration of snow-related parameters in hydrological models (Sun et al., 2019) and the hydrologic design in snow-dominated environments (Yan et al., 2018). Studies on the impact of climate variability and change on snowpack evolution can also rely on snow measurements from national networks (e.g., Clark et al, 2001; Musselman et al., 2017). Manual snow surveys and automatic SWE stations with collocated snow depth (SD) measurements can provide information on the bulk density of the snowpack. These data have been used to develop and evaluate methods to estimate bulk snow density from snow depth and different predictors (e.g., Sturm et al., 2010; Hill et al., 2019; Ntokas et al., 2021) and to correct biases in large-scale gridded SWE products (Pulliainen et al., 2020).

Snow covers almost 85% of Canada's landmass during winter (December-March mean monthly snow cover extent for 1976 – 2019: 8.40 million km$^2$; ECCC, 2020). In Canada, the vast majority of in situ SWE measurements are collected by provincial or territorial governments and hydropower companies. Despite the importance of these measurements for pan-Canadian applications in hydrology, climate monitoring and applied research, there is no central agency tasked with the ongoing coordination, maintenance, and archiving data collected from these various agencies. SWE is not measured by the pan-Canadian network of manual and automatic stations operated by Environment and Climate Change Canada (ECCC), except at select stations in northern Canada. ECCC manual and automatic stations only report SD (Brown et al., 2021). Historically, the Government of Canada's Atmospheric Environment Service (AES, now the Meteorological Service of Canada (MSC), part of ECCC) coordinated the reporting and archiving of snow survey data from various agencies (including AES) between 1955 and 1985 in the form of yearly Snow Cover Data (SCD) bulletins (Braaten 1998). Since the mid 1980s, there has been no ongoing coordinated effort to archive snow survey data from various reporting agencies across Canada. The Canadian Historical Snow Survey dataset (CHSSD) was borne out of a data recovery effort of the mid 1990s, led by AES, which aimed to digitize the AES SCD books and combine it with available data from other agencies. This digital dataset, which was released in 2000, combined seven datasets from six different agencies (Braaten 1998). Methods and quality control procedures are outlined in Braaten (1998). This database was updated for the first time in 2004 (Hill, 2004). The most recent update, released in 2019 (Brown et al., 2019), contained data up to and including the 2016/17 snow season. It is referred in the rest of the text as the 2019 CHSSD update. With each database update, some agencies (and sites) are added while others are not updated. The 2019 update included new sites in Yukon Territory, Northwest Territories, British Columbia and northern Manitoba. Some regions, such as Saskatchewan, Newfoundland and Labrador, and Quebec were not updated, either because a data custodian could not be identified or because an agency ceased snow survey operations or did not allow data sharing.

The 2019 CHSSD update has been used in numerous studies (see Table A1 for a complete list). However, researchers working with the 2019 CHSSD update have reported a number of errors in metadata (e.g., incorrect snow survey coordinates and elevations) and the presence of a large amount of duplicate data. These issues, combined with the need for coordinated regular updates of in situ SWE observations, highlighted a need for a reworking of the CHSSD. The objective of this paper is to provide a detailed description of the development of the Canadian historical SWE dataset (CanSWE), which replaces the CHSSD. The dataset name was changed to reflect the inclusion of automated SWE data and to highlight SWE as the dataset's primary variable of interest. The methodology presented here will serve as a basis for future regular and coordinated updates of the CanSWE dataset. The paper is organized as follows. Section 2 describes the different steps involved in creating the CanSWE dataset, including quality control. Section 3 gives an overview of the spatial and temporal coverage of the dataset and provides details on the data and metadata included in this dataset. Finally, Sect. 4 describes the data availability and Sect. 5 offers concluding remarks and perspectives about future updates of CanSWE.

## 2 Creation of the CanSWE dataset

The creation of the new Canadian historical SWE (CanSWE) dataset from the most recent version of the CHSSD involved three main steps as detailed on Fig. 1: (i) correction and cleaning of the 2019 CHSSD update, (ii) update of this cleaned dataset to July 2020 and addition of snow data from new stations and agencies, and (iii) consistent quality control (QC) of the final dataset. These steps are described in the next sections.

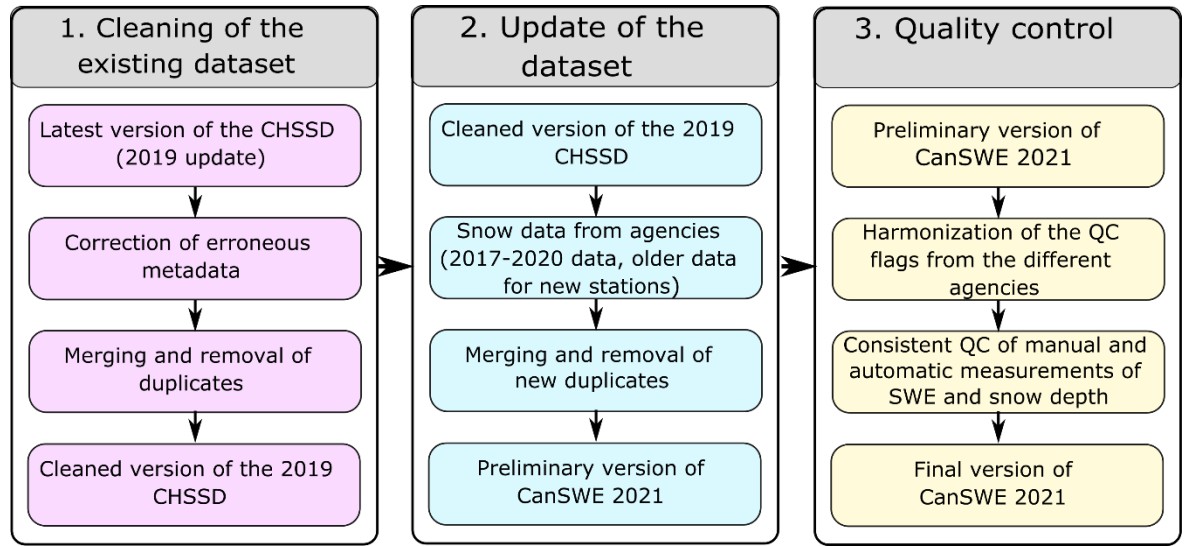

**Figure 1: CanSWE dataset creation workflow.**

### 2.1 Cleaning of the 2019 CHSSD update

#### 2.1.1 Correction of erroneous metadata

The 2019 CHSSD update released by Brown et al. (2019) contained snow data from 3124 individual stations across Canada. Prior to adding new data, the existing data were scrutinized to identify and resolve several issues raised by researchers working with the 2019 update. A preliminary analysis consisted of identifying stations with erroneous or incomplete metadata: (i) blank station name, (ii) placeholder text for station name, (iii) missing latitude and/or longitude, (iv) obvious errors in latitude and/or longitude and/or elevation. A total of 91 stations were identified and were manually checked. Valid data for station name and/or coordinates were obtained from databases of the originating agencies for 28 stations and the corresponding changes were made to the CHSSD. The remaining 63 stations with erroneous/incomplete metadata and their corresponding records of snow data were excluded, leaving 3061 individual stations in the dataset.

#### 2.1.2 Merging and removal of duplicates

A second analysis was then carried out to remove duplicates and improve the consistency of the database prior to adding any new data. Duplicates are defined as stations with different station IDs and potentially with different metadata (station name

and/or coordinates and/or elevation) having the same SWE observation for multiple dates (at least ten). Duplicates usually consist of a pair of stations but can also be formed of three or four stations. Duplicates were introduced in previous updates of the CHSSD when snow data from various agencies were added to the CHSSD without ensuring that incoming data were already present in the CHSSD under a different station ID. In particular, instances of data duplication were introduced when the SCD books were digitized. Stations from these books were all assigned a unique ID (station with the prefix "SCD-") which differs from that of the agency of origin. This generated a substantial amount of duplicate data during the period 1956 – 1986. Duplicates were also introduced in transboundary situations where a single station is archived by multiple agencies but under different station names and IDs.

Duplicates were identified through a combination of automated station selection and manual inspection. For each station in the CHSSD (referred here as 'inspected station'), all stations within a 5-km radius were identified. Each group of neighbouring stations was then manually inspected for similarities in (i) snow measurements for matching dates (at least ten), (ii) station location and (iii) station name. In most cases, all three of the criteria were satisfied to trigger a decision on whether a duplicate was identified. When a duplicate was identified, the inspected station and its matching neighbours were assigned a unique merging key to be used in subsequent consolidation. If no similar stations to the inspected station were identified in a group of neighbouring stations, the inspected station was assigned its own merging key to aid in future updates to the CHSSD. Isolated stations without neighbours in a 5-km radius and without having been assigned a merging key were then inspected. For these isolated stations, the five nearest stations – regardless of distance – were identified, and the same similarity criteria were applied within each group of stations. As before, a unique merging key was assigned to each set of identified duplicate stations, or only to the reference station itself in the case of no duplicates being identified. As a final check, for each station, a query over the full list of station names was carried out using a shortened version of the station name to identify stations in the CHSSD with similar names. These stations were then manually inspected for similarities as described above. In total, 842 groups of duplicate sites were identified: among them, 788 were comprised of two stations, 52 had three stations and two had four stations.

The final step consisted of removing the duplicates. For each merging key associated with a set of duplicate stations, a single reference station ID was identified. When duplicates occurred between one or several IDs from the SCD books and an ID from an originating agency, the reference ID was taken as that of the originating agency. When duplicates occurred between IDs from several agencies (typical of transboundary situations), the station ID belonging to the provincial or territorial agency where the station is located was selected as the reference ID. Finally, when duplicates occurred between IDs in the SCD books or IDs from the same agency, the ID associated with the longest SWE record was selected as the reference ID. Records of snow depth and SWE from the reference station were retained and records from the duplicate stations inserted on dates when no data were present in the records from the reference station. The metadata (coordinates and elevation) were taken from the reference station. The station IDs and names of the duplicate sites were retained as alternative IDs and names to facilitate future data enquiries using IDs and names present in the previous versions of the CHSSD. The duplicates'

metadata and data were then removed from the CHSSD, for a total of 898 stations removed. Duplicated data were mostly
removed over the period 1956-1986 (Fig. 2) due to conflicts between the data from the SCD books and the data from the
agency of origin. The cleaned version of the CHSSD contains 2163 individual stations and was used as the basis for the
update presented in this paper.

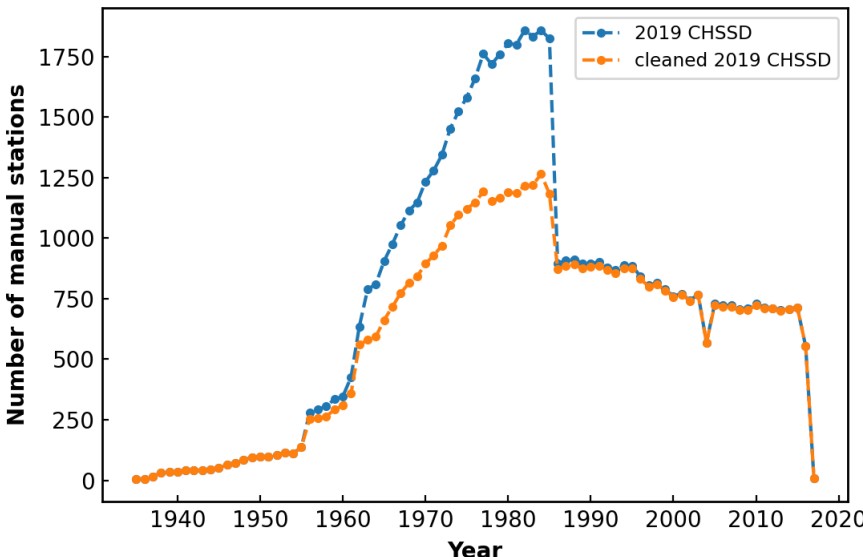

**Figure 2: Number of manual snow survey sites reporting at least one measurement between 1 February and 30 April in the**
155 **original 2019 CHSSD update before (blue) and after (orange) the removal of duplicate stations.**

### 2.2 Update of the CHSSD

Agencies collecting SWE measurements across Canada were contacted to obtain access to snow data (SWE and SD). Table
1 lists the twelve different agencies that contributed snow data to the update leading to the CanSWE dataset. These agencies
correspond to provincial and territorial agencies responsible for streamflow forecasting and/or environmental monitoring and
160 hydropower companies. All Canadian provinces and territories are covered by this update, with the exception of Nova Scotia
and Prince Edward Island where no snow measurement program is currently active at the provincial level. Nunavut is
included through the manual snow survey data collected at stations managed by ECCC. Snow survey data were also
provided by the Government of Manitoba, but their format precluded inclusion in CanSWE at this time.

**Table 1: Agencies that provided snow measurements used in this study. The table makes the distinction between manual and automatic snow measurement stations.** *Updated* **stations correspond to stations already present in the 2019 CHSSD update for which data for the recent years (2017-2020) have been added whereas** *new* **stations were not present in the 2019 update of the CHSSD.**

| Agency | Manual stations | | Automatic stations | |
|---|---|---|---|---|
| | Updated | New | Updated | New |
| Yukon Water Resources Branch | 56 | 1 | 0 | 0 |
| Government of Northwest Territories | 47 | 24 | 0 | 0 |
| Meteorological Service of Canada (ECCC) | 14 | 0 | 0 | 0 |
| British Columbia Ministry of Environment | 160 | 1 | 55 | 35 |
| Alberta Environment and Parks | 112 | 1 | 0 | 16 |
| Saskatchewan Water Security Agency | 0 | 172 | 0 | 0 |
| Manitoba Hydro | 24 | 11 | 0 | 0 |
| Ontario Power Generation | 42 | 0 | 0 | 0 |
| Ontario Ministry of Natural Resources and Forestry | 232 | 8 | 0 | 0 |
| Hydro Québec | 0 | 80 | 0 | 64 |
| Government of New Brunswick | 56 | 2 | 0 | 0 |
| Government of Newfoundland and Labrador | 0 | 25 | 0 | 4 |
| Total | 743 | 325 | 55 | 119 |

The snow data provided by the different agencies consist of two types of measurements: (i) manual gravimetric snow surveys and (ii) automatic stations. Manual snow survey data were provided by the twelve originating agencies (Table 1). These data are collected by field observers using snow corers typically at five to ten points along a pre-determined survey line of 150-300 m selected to be representative of the land cover and terrain, although the precise methodology varies by agency (Brown et al., 2019). Manual snow surveys are collected irregularly in time and the sampling frequency varies from one agency to another (Fig. 3). A majority of agencies conduct snow surveys once or twice per month during the snow season (Fig 3b) but several (e.g., Saskatchewan, Newfoundland and Labrador, Northwest Territories, Fig. 3a) only conduct measurements close to the peak snow accumulation and during the melting period for hydrological purposes. Most of the agencies use the Federal snow sampler whereas the Prairie and the ESC-30 samplers are used in regions of shallow snowpack such as the Prairies or the Arctic (Table 2). The Federal snow sampler is a small-diameter and multi-section sampler design to aid sampling in deep snowpack whereas the Prairie and the ESC-30 samplers present large diameters tubes to maximize snow collection in shallow snow cover and increase measurement accuracy (Dixon and Boon, 2012). More details about the impact of sampler type on uncertainties in SWE measurements are given in Goodison et al. (1987) and Lopez Moreno et al. (2020). Automatic SWE measurements from snow pillows were provided by the British Columbia

Ministry of Environment (hourly measurements) and Alberta Environment and Parks (daily measurements) (Table 2). Hydro Quebec and the Government of Newfoundland and Labrador also provided hourly automatic SWE measurements from passive gamma radiation sensors (Choquette et al., 2013; Table 2). Most of these automatic stations are also equipped with automatic measurements of snow depth using ultrasonic ranging instruments.

**Table 2: Equipement for manual (snow samplers) and automatic SWE measurements used by each agency that provided snow measurements for CanSWE.**

| Agency | Manual stations | Automatic stations |
|---|---|---|
| Yukon Water Resources Branch | Federal sampler | - |
| Government of Northwest Territories | ESC-30 sampler | - |
| Meteorological Service of Canada (ECCC) | ESC-30 sampler | - |
| British Columbia Ministry of Environment | Federal sampler | Snow pillows |
| Alberta Environment and Parks | Federal sampler | Snow pillows |
| Saskatchewan Water Security Agency | Prairie sampler | - |
| Manitoba Hydro | Federal sampler | - |
| Ontario Power Generation | Federal sampler (at most sites), ESC-30 sampler at some sites | - |
| Ontario Ministry of Natural Resources and Forestry | | - |
| Hydro Québec | Federal sampler | Passive gamma radiation sensors |
| Government of New Brunswick | Federal sampler | - |
| Government of Newfoundland and Labrador | Federal sampler | Passive gamma radiation sensors |

The snow data and the corresponding metadata from the different agencies were obtained by direct download from web pages or ftp servers, from requests on web data servers, or directly via email. Data were most often provided as csv or Excel files but were also received as text bulletins, zxrp files, and ESRI Shapefiles. Python routines specific to each agency and corresponding data format were written to process the data and metadata and arrange them in a consistent NetCDF format. Snow depth and SWE data were included at a daily frequency. Hourly time series from automatic stations were first pre-processed with a 24-h median filter to remove noise (Stone, 1995), especially in the snow depth time series from ultra-sonic sensors. The filtered data corresponding to 18 UTC was then extracted from the hourly time series to get a daily value. 18 UTC was selected since it corresponds to daytime in Canada. When available, the quality control flags from the originating

agency were added (see Sect. 3.3 for more details on QC). Finally, a station metadata record was constructed for each snow survey site including station ID, data source agency, station name, latitude, longitude and elevation. This list of metadata variables corresponds to that used in the 2019 CHSSD update (Brown et al., 2019). When elevation was not present in the metadata from the originating agency it was extracted from the United States Geological Survey's National Elevation Dataset (USGS NED, Gesh et al., 2002) at the position corresponding to the location of the snow survey site. The USGS's NED covers all Norther America at 30-m resolution (except parts of Alaska) and has a vertical accuracy of 3.53 m over Canada (Gesch et al., 2014). A new code was also added in the metadata to describe the method of SWE measurements at each snow survey site. This code follows the standards of the World Meteorological Organization (WMO, 2019a) and is described in Table 3. Information about the sitting of the snow measurement sites (e.g., open terrain, below forest, clearing) is not available in the present version of CanSWE and will be added to future version of the dataset.

**Table 3: WMO SWE measurement codes (WMO, 2019a)**

| Code | Method of SWE measurement |
|---|---|
| 0 | Multi point manual snow survey |
| 1 | Single point manual SWE measurement |
| 2 | Snow pillow or snow scale |
| 3 | Passive gamma |
| 4 | Global Navigation Satellite System/ Global Positioning System methods |
| 5 | Cosmic ray attenuation |
| 6 | Time domain reflectometry |

As a last step, snow data from the different agencies and the corresponding metadata were added to the NetCDF file containing the cleaned 2019 CHSSD update (Sect. 2.1). For stations already present in this file, the new snow data (from the beginning winter of 2016-2017 to the end of July 2020) were simply appended to the existing time series. Data from new snow survey sites were also added (Table 1). They consisted of newly established snow survey sites over the period 2017-2020 and of historical snow survey sites that were not included in the 2019 CHSSD update. For example, historical manual snow survey data were added from Hydro Quebec, the Saskatchewan Water Agency, the Governments of Northwest Territories and the Government of Newfoundland and Labrador. The full historical archive of the snow pillow data from Alberta Environment and Parks was also added to CanSWE. Finally, new data from automated passive gamma radiation sensors from Hydro Quebec and the Government of Newfoundland and Labrador were added. This is significant because no data from automatic stations from Eastern Canada were present in any previous version of the CHSSD. Duplicates created by the addition of new stations were identified and removed following the methods described in Sect. 2.1.2. Overall, 798 stations from the cleaned 2019 CHSSD update were updated to 2020 and 444 new stations were added. The CanSWE dataset

contains snow data for 2607 sites across Canada (Table 1). Finally, where both SWE and SD measurements were available, bulk snow density was calculated from the ratio of SWE to SD and included in the final database.

**2.3 Quality control of the final dataset**

Quality control of CanSWE involved two main steps: (i) homogenization of data quality flags from the various reporting agencies, (ii) QC of the manual and automated SWE and SD records. Each of the twelve reporting agencies have their own data archiving and reporting system with many agencies using data flags to identify possibly erroneous or problematic measurements. For example, it is not always possible to accurately measure trace amounts of snow or to estimate SWE in
patchy snow conditions. In these instances, the measurement may be reported as 0 but a flag of T (trace) or P (patches) assigned. Most, but not all, agencies conduct their own internal quality control prior to releasing their data. Instances where data have been revised by the originating agency are often flagged, as are cases when the originating agency estimated the SWE or SD value, or when problems were encountered during sampling. It is important to note that not all agencies use internal data flags and not all agencies flag the same types of issues. For example, snow patches are only reported by four
originating agencies and trace amounts of snow are reported by eight.

The publicly released dataset of Brown et al. (2019) did not include agency flags. This information is an important addition to CanSWE. For each agency, we identified all existing flag values and their respective definitions. This process highlighted two key issues: (i) the same flag value had a different meaning depending on the reporting agency and/or type of measurement, (ii) the same meaning was represented by different flag values depending on the reporting agency and/or type
of measurement. A conversion table was created to reassign flag values from the various agencies into a single set of standard values and definitions. New flag values were added where necessary. The final dataset contains 10 and 8 agency flags for SWE (data_flag_snw) and SD (data_flag_snd) (Table 4), respectively, compared to 18 and 15 before homogenization.

 **Table 4: Agency data flags used in CanSWE (see Sect.2.3)**

| Data flag | Definition |
|---|---|
| A | Sampling problems |
| B | Manual snow survey conducted outside the nominal sampling period. |
| C | Combination of A and B |
| E | Estimate |
| G | Measurement location >1 km from station coordinates. This flag is specific to manual snow survey data provided by the Saskatchewan Water Security Agency beginning in 2011. |
| M | Missing |
| P | Patches |
| R | Revised data |
| T | Trace |
| Y | Precise sampling date not available - set to April 1st. Flag is specific to manual snow survey data provided by the Government of the Northwest Territories. |

Quality control (QC) of SWE and SD measurements included range thresholding and automated outlier detection. SWE and SD QC flag variables (qc_snw and qc_snd, respectively), which are separate and distinct from the agency flag variables were added to the dataset (Table 5). The set of QC procedures implemented here is self-contained, applicable to the full dataset, and does not rely on any auxiliary data. Researchers using a subset of CanSWE for a local region or specific years may wish to conduct their own independent QC that may consider available temperature and precipitation information (e.g., Johnson and Marks, 2004; Yan et al., 2018).

**Table 5: QC flags used in CanSWE (see Sect. 2.3)**

| QC flag | Definition |
|---|---|
| H | SD > 3 m (>8 m west of -113° longitude). SD set to *NaN* |
| M | Data masked (set to *NaN*) in a previous CHSSD update |
| V | Automatic SD-SWE measurement identified as outlier using robust Mahalanobis[a] distance. SD and SWE set to *NaN* |
| W | SWE > 3000 kg m$^{-2}$ (>8000 kg m$^{-2}$ west of -113° longitude). SWE set to *NaN* |
| D | Derived bulk snow density failed 25 - 700 kg m$^{-3}$ threshold. SD, SWE and derived bulk snow density set to *NaN* |

[a] See Leys et al. (2018) for more details

Range thresholds were used to identify spurious records in both automated and manual measurements. Brown et al. (2019) applied this method to remove outliers from the 2019 CHSSD update and only keep valid triplets of SWE, SD and bulk snow

density. For CanSWE, we adopted the thresholds outlined in Brown et al. (2019) for SWE and SD (0 – 3000 kg m$^{-2}$, 0 – 8000 kg m$^{-2}$ for mountain) but a slightly more restrictive range of 25 – 700 kg m$^{-3}$ (as opposed to 50 – 1000 kg m$^{-3}$) for bulk
snow density. These ranges are based on common ranges for SWE and SD from the literature (see Braaten, 1998). The range thresholding applied to bulk snow density aims at identifying SWE-SD pairs that are likely erroneous. To maintain consistency of the long-term database we used the same definition for *mountain* as Brown et al. (2019) where *mountain* is defined as all land west of -113° longitude. This definition is very simple and more advanced definitions (e.g., Karagulle et al,, 2017) may be considered in future version of CanSWE. Measurements outside the specified ranges were set to *NaN* and
QC flags assigned according to Table 5. When a record failed the SWE (SD) threshold but not the SD (SWE) threshold only the SWE (SD) value was set to *NaN*; the corresponding density value was also set to *NaN* and a *W* or *H* flag assigned to these records (Table 5). When a record failed the bulk snow density threshold SWE, SD and bulk snow density were set to *NaN*; a *D* flag was assigned to these records (Table 5). Together, these steps masked one or both of SWE and SD in 0.17% and 5.5% of the manual and automated records, respectively. Table 6 lists the number and percentage of records masked at
each QC step. The available data before and after QC is shown in Fig. 3. The small number of records flagged using the range thresholds is not surprising given that much of the data underwent QC in previous updates. The SWE and SD ranges are unchanged from previous updates so only data added in the current update have the possibility of being flagged. The density range is slightly more conservative so both new and old data were removed. Consequently, the density range flagged the most records when compared to the SWE and SD thresholds. Finally, when SWE (SD) measurements were masked (set
to *NaN* ) in previous CHSSD updates for any reason, the corresponding QC flag (qc_snw/qc_snd) was set to *M* (missing) in CanSWE. 0.3% and 1.6% of the manual and automated records, respectively, have *M* flags.

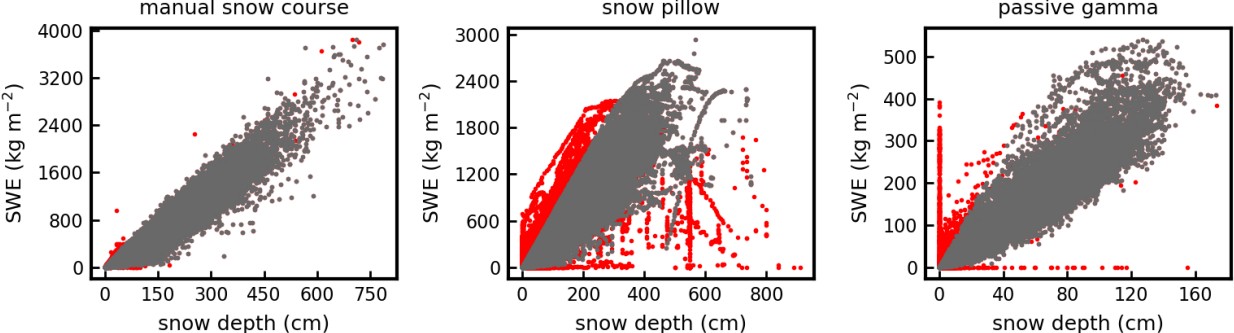

**Figure 3: CanSWE data by measurement type before (red) and after (grey) quality control described in Section 2.3 and Table 65. Snow pillows are deployed in British Columbia and Alberta; passive gamma radiation sensors are used by Hydro Quebec and the**
285 **Government of Newfoundland and Labrador (Sect. 2.2).**

Table 6: Number of manual and automated records masked (set to *NaN*) at each quality control step. Percentage relative to final dataset that has 1,072,229 records: 312,551 manual and 759,678 automated.

| QC step | QC flag | Number of records flagged | | % of total | |
|---|---|---|---|---|---|
| | | manual | auto | manual | auto |
| SD threshold: SD > 3 m, 8 m west of 113°W | H | 1 | 343 | <0.01% | 0.05% |
| SWE threshold: SWE > 3000 kg m$^{-2}$, 8000 kg m$^{-2}$ west of 113°W | W | 7 | 4405 | <0.01% | 0.59% |
| Derived bulk snow density threshold: 25 - 700 kg m$^{-3}$ | D | 517 | 37140 | 0.17% | 5.0% |
| RMD threshold (Sect. 2.3) | V | n/a | 1177 | n/a | 0.16% |
| Data masked in previous CHSSD updates | M | 824 | 12156 | 0.26% | 1.62% |

Additional quality control measures were applied to the automated data but was applied to the manual data due to their low temporal sampling frequency. We used the robust sample Mahalanobis distance (RMD) (Leys et al. 2018) to identify spurious SWE – SD data pairs as in Hill et al (2019). The RMD method is based on the traditional Mahalanobis Distance (MD) (Mahalanobis, 1930), which is the distance of a point from the mean of a multivariate distribution. It relies on the mean and covariance matrices of the multivariate distribution, which are affected by outliers. The RMD uses the Minimum Covariance Determinant (Rousseeuw, 1984) and is less sensitive to outliers than the MD (Leys et al., 2018). Because this method relies on a multivariate dataset only automated data with both SWE and SD observations were assessed. For each site with a minimum of 20 records, the RMD was calculated for each SWE – SD data pair. Following Hill et al. (2019), outliers were defined as a square RMD larger than the upper 0.001 quantile of a chi-squared distribution with $p$ degrees of freedom ($X^2_p$; where $p$ = number of dimensions of the data) (Gnanadesikan and Kettenring, 1972). For these records SWE, SD, and density were set to *NaN* and QC flags (qc_flag snw, qc_flag_snd) assigned *V* (Table 4). This step masked an additional 0.16% of automated records.

# 3 Spatial and temporal coverage of the final dataset

Figure 4 shows the location of the 2607 sites included in the CanSWE dataset. It highlights the concentration of observations in the southern populated regions of Canada. The majority of the manual data are from Ontario and British Columbia (Fig. 5). Importantly, there are large data gaps in Nunavut and in the northern regions of Quebec, Ontario and Saskatchewan. The update of historical data in Yukon and the Northwest Territories and the establishment of new sites in the Northwest

Territories improved the spatial and temporal coverage of CanSWE in the western part of the Canadian Arctic compared to

the 2019 CHSSD update. A few snow survey sites are found in the USA close to the border with Alberta and British Columbia. These sites are in the headwater catchments of rivers flowing into Canada. Similarly, data from northern parts of the USA state of Maine are included in the data from New Brunswick.

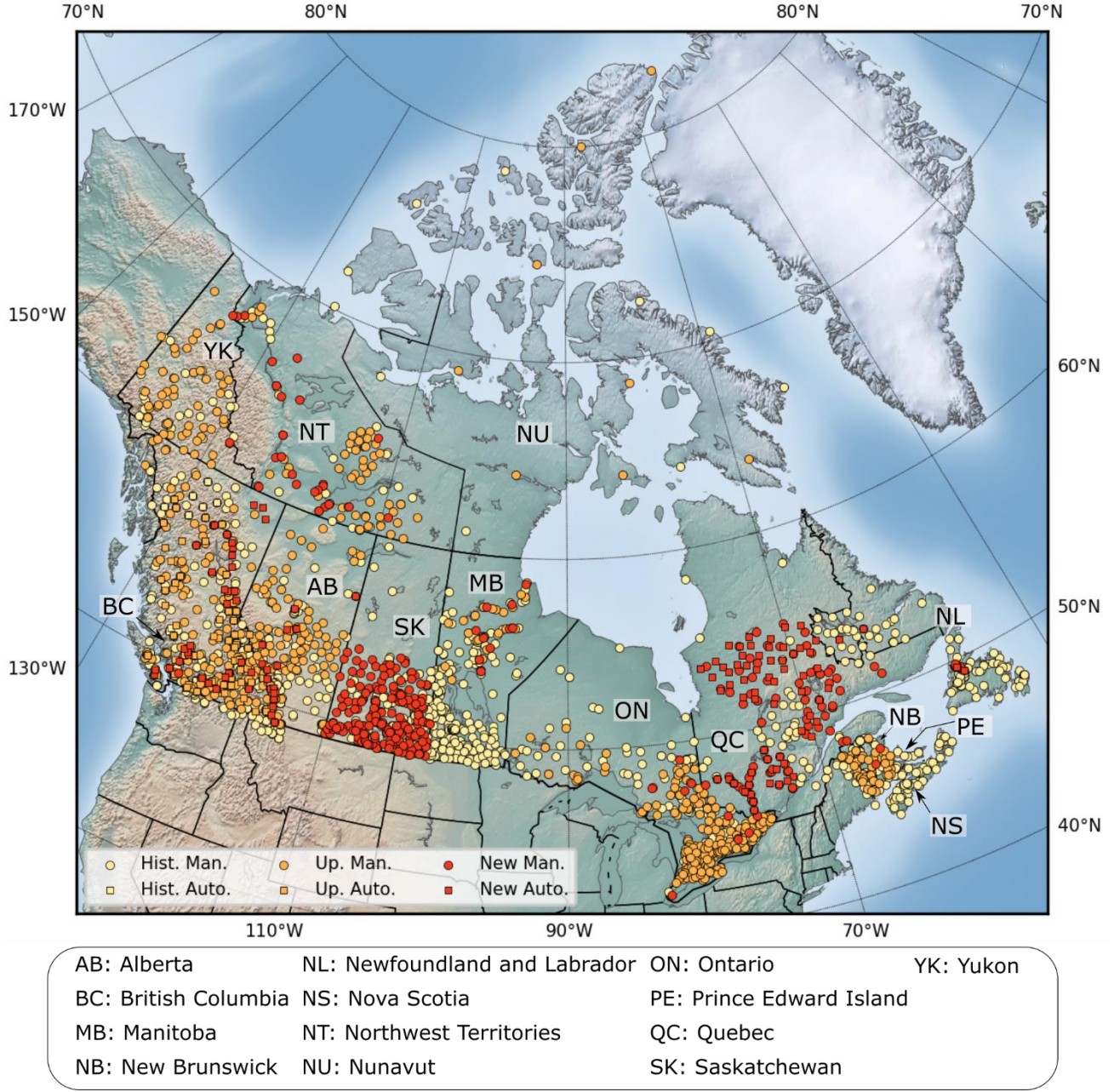

| | | | |
|---|---|---|---|
| AB: Alberta | NL: Newfoundland and Labrador | ON: Ontario | YK: Yukon |
| BC: British Columbia | NS: Nova Scotia | PE: Prince Edward Island | |
| MB: Manitoba | NT: Northwest Territories | QC: Quebec | |
| NB: New Brunswick | NU: Nunavut | SK: Saskatchewan | |

**Figure 4: Snow measurements sites (manual and automatic) contained in CanSWE. The distinction is made between new historical sites added during this update (New), those (Updated (Up.)) present in the 2019 CHSSD update for which 2017-2020 snow data have been added and those (Historical (Hist.)) present in the 2019 CHSSD update for which no data have been added.**

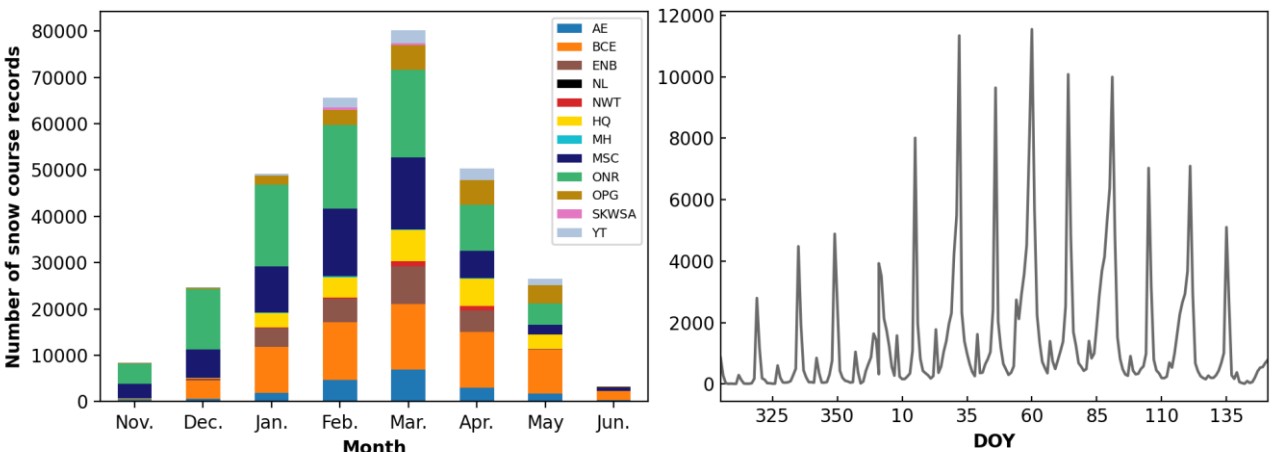

**Figure** 5**: Number of manual snow survey records by contributing agency and month (left) and by day of year (right) between 1991 and 2020. AE: Alberta Environment and Parks, BCE: British Columbia Ministry of Environment, ENB: Government of New Brunswick, NL: Government of Newfoundland and Labrador, NWT: Government of Northwest Territories, HQ: Hydro Quebec and partners, MH: Manitoba Hydro, MSC: Meteorological Service of Canada (ECCC) and observations previously conducted by now Crown-Indigenous Relations and Northern Affairs Canada, ONR: Ontario Ministry of Natural Resources and Forestry, OPG: Ontario Power Generation, SKWSA: Saskatchewan Water Security Agency, YT: Yukon Water Resources Branch.**

Figure 6 compares the distribution of the station elevation with the hypsometry in each province and territory. The hypsometry has been derived from the Global Multi-resolution Terrain Elevation Data 2010 (https://www.usgs.gov/core-science-systems/eros/coastal-changes-and-impacts/gmted2010, last access 21 July 2020) at 30 arc-seconds reprojected to the Canada Albers Equal Area Conic projection at 250-m grid spacing. Figure 6 shows that the elevation coverage provided by the stations varies greatly from one region to another. A representative coverage is found in provinces of Eastern Canada (Quebec, New Brunswick, Nova Scotia). On the other hand, in British Columbia and Alberta, SWE measurement sites tend to be located at higher elevation than the average terrain to provide relevant information on snow cover in mountainous headwater catchments. Large differences between the station elevation coverage and the hypsometry are also found in Nunavut and Saskatchewan. They are associated with sparse spatial coverage in the elevated inland parts of Nunavut and in the low-elevation northern part of the province in Saskatchewan.

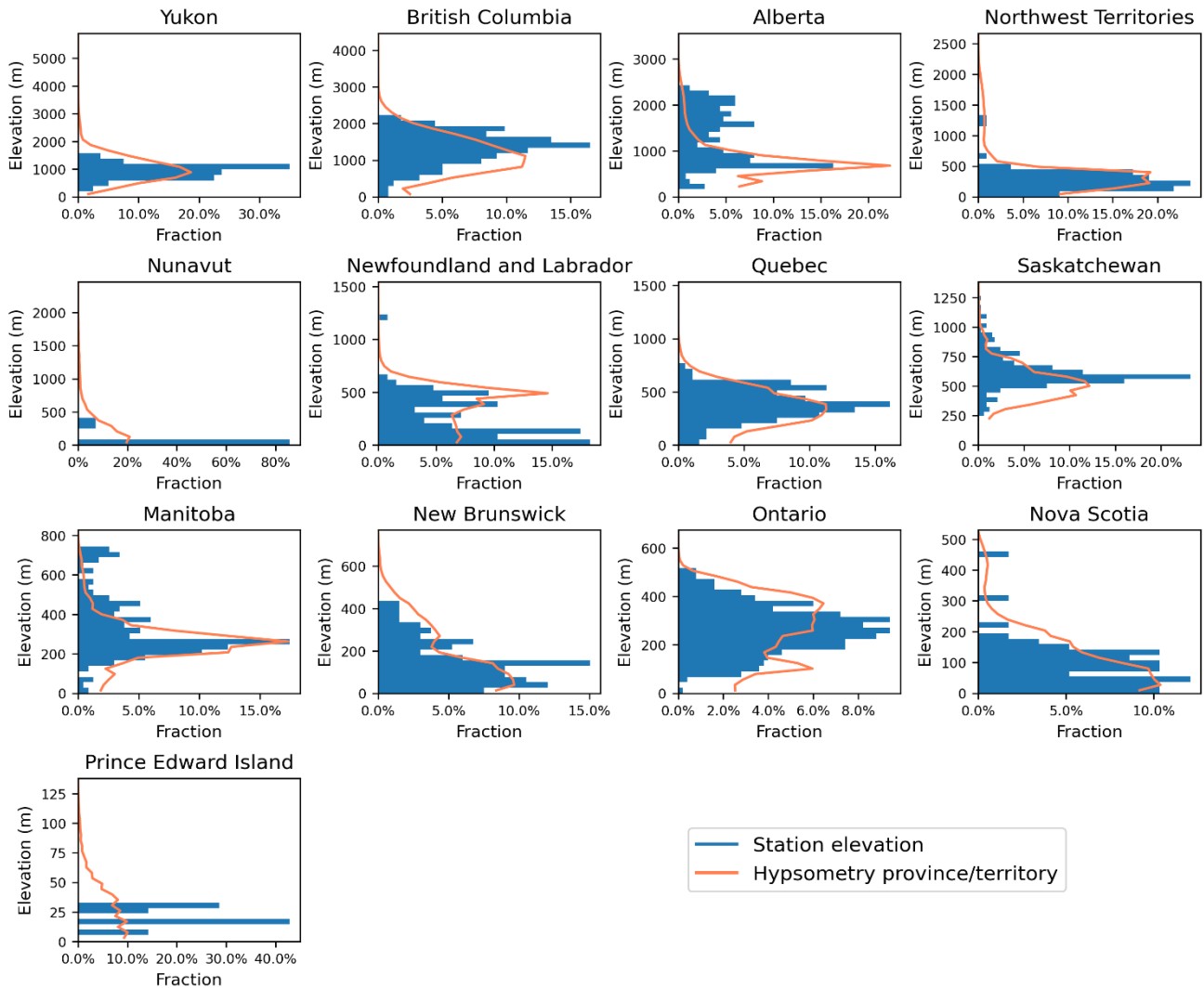

**Figure 6: Distribution of the station elevation and terrain elevation for each province and territory. Note the changing maximal values on the y-axis of the different sub-figures.**

Figure 7 displays the temporal distribution of number of reporting stations in CanSWE by province and territory. SWE data are available over the period 1928 – 2020. Across Canada, the maximum number of stations was reached in 1984 with 1288 stations reporting at least one SWE measurement for this snow season. The strong decrease in number of stations after 1985 is due in part to cessation of the publication of the coordinated yearly Snow Cover Data bulletins by ECCC (see Sect. 2 for more details). The availability of data from provinces such as Manitoba, Saskatchewan and Newfoundland and Labrador were strongly impacted by the end of this coordination effort. The addition of snow course data from the Saskatchewan Water Security Agency in CanSWE (Table 1) improved the availability of snow data for the more recent years in this province. Ontario and British Columbia have the largest number of snow survey sites.

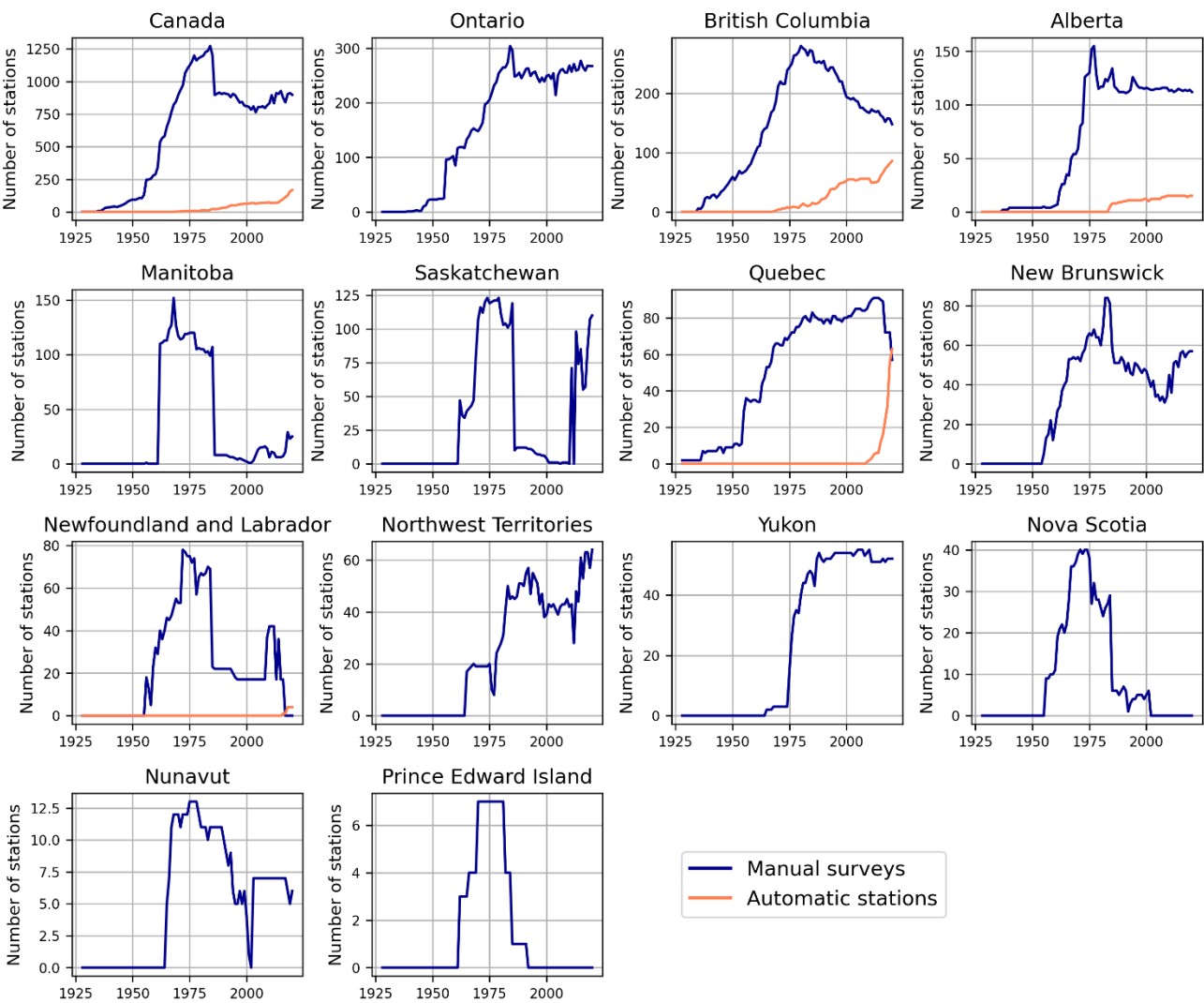

**Figure 7: Evolution of the number of stations reporting SWE measurement per snow season for Canada (upper left) and each province and territory. A snow season corresponding to year Y, is defined as starting on September 1st of year Y-1 and ending on August 31st of year Y. Note the changing maximal values on the y-axis of the different sub-figures.**

The first automatic stations measuring SWE (snow pillows) in Western Canada were deployed in British Columba in the late
60s and early 70s. In Eastern Canada, the installation of automatic GMON sensors is more recent and started in 2009 in Quebec. In the CanSWE dataset, measurements from automatic stations first outnumbered those from manual snow surveys in 1988 and accounted for 89% of total SWE records for snow season 2020 (Fig. 8). The higher proportion of automated data is largely due to their higher measurement frequency compared to manual snow surveys. Finally, the number and frequency of manual snow survey observations varies over the course of the snow season and between reporting agencies (Fig. 5). The
number of snow surveys increases over the accumulation season reaching a maximum during the period of peak snow

accumulation with February and March having the highest numbers of manual snow surveys. Peak SWE occurs later in the northern regions and in mountainous regions, but the seasonal peak shown in Fig. 5 reflects the concentration of observations in southern Canada.

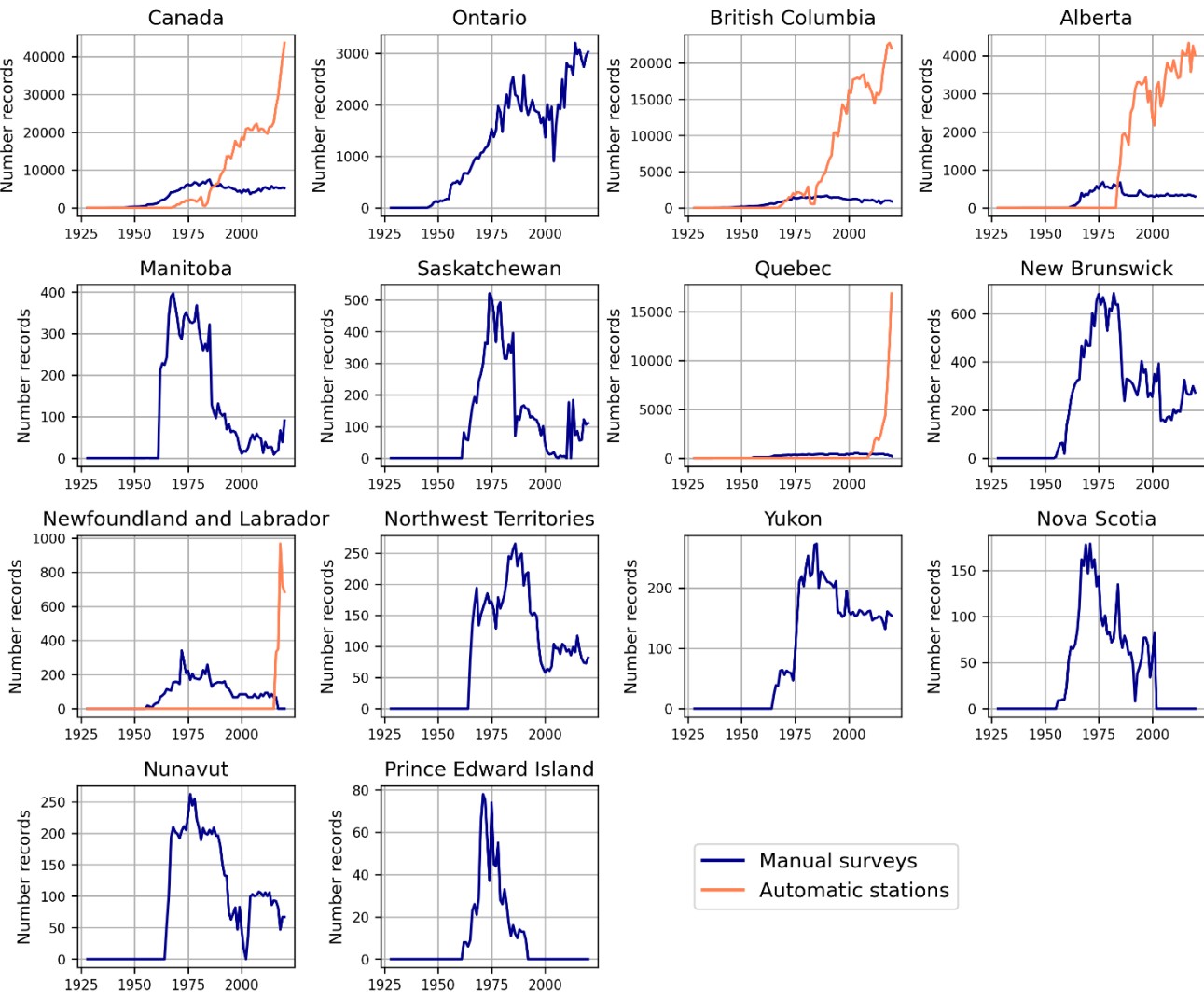

Figure 8: Same as Fig. 7 for the number of SWE records per snow season. The order of the provinces and territories is the same as in Fig. 6. Note the changing maximal values on the y-axis of the different sub-figures.

## 4 Data availability

The CanSWE dataset is distributed as a single file in NetCDF format that follows the Climate and Forecasts (CF) Metadata Conventions (Hassel et al., 2017). It is available at https://doi.org/10.5281/zenodo.4734372 (Vionnet et al., 2021). Table 7 describes the data and observational metadata contained in this file. Readme files in English and French are also included in

the Zenodo data repository. Future versions of CanSWE will include updated names for the observational metadata to follow the WMO standards (WMO, 2019b)

**Table 7: Description of the variables (dimensions, metadata, data and quality-control flags) present in the netcdf file containing the CanSWE dataset**

| Type of variable | Variable name | Description | Dimension | Units |
|---|---|---|---|---|
| Dimension | station_id | Station identification code | station_id | (-) |
| | time | Time | time | day |
| | lat | Station latitude | station_id | deg. north |
| | lon | Station longitude | station_id | deg. east |
| | elevation | Station elevation | station_id | m |
| | source | Data provider | station_id | (-) |
| Observational metadata | station_name | Primary station name | station_id | (-) |
| | station_name_sec | Secondary station name | station_id | (-) |
| | station_name_ter | Tertiary station name | station_id | (-) |
| | station_id_sec | Secondary station identification code | station_id | (-) |
| | station_id_ter | Tertiary station identification code | station_id | (-) |
| | type_mes | Method of measurement for SWE[1] | station_id | (-) |
| Data | snw | Water equivalent of snow cover (SWE) | station_id, time | kg m$^{-2}$ |
| | snd | Snow depth (SD) | station_id, time | m |
| | den | Bulk snowulk density | station_id, time | kg m$^{-3}$ |
| Quality-control flag | data_flag_snw | Agency data quality flag for SWE[2] | station_id, time | (-) |
| | data_flag_snd | Agency data quality flag for SD[2] | station_id, time | (-) |
| | qc_flag_snw | CanSWE quality control flag for SWE[3] | station_id, time | (-) |
| | qc_flag_snd | CanSWE quality control flag for SD[3] | station_id, time | (-) |

[1] see Table 3 for more details; [2] see Table 4 for more details, [3] see Table 5 for more details.

**5 Conclusion**

The Canadian historical SWE dataset (CanSWE) contains measurements of water equivalent of snow cover (SWE) and snow depth (SD) and bulk snow density for an ensemble of sites across Canada. This dataset includes the results of extensive cleaning and quality control of the existing Canadian Historical Snow Survey Dataset (CHSSD), the addition of new historical data sources, and an update to July 2020 with data from 12 organizations and their partners. New stations from Hydro Quebec, the government of Newfoundland and Labrador, the government of Northwest Territories and the

Saskatchewan Water Security Agency were added and improved the spatial coverage. A systematic quality control was applied to identify and remove outliers in SWE, SD, and bulk snow density. The CanSWE dataset presented in this paper includes data from 2607 manual and automatic snow survey sites across Canada over the period 1928 – 2020. We anticipate that these data will be used for (i) climate monitoring and research, (ii) evaluation of land surface and hydrological models, (iii) development and evaluation of snow products, and (iv) other snow-related activities. Regular updates are required to make such datasets useful for the community. Ideally, these updates should be carried out on a yearly basis at the end of each snow season. The data ingestion routines and automated quality control procedures developed under this project will allow future updates to be carried out in a timely and systematic fashion. We also hope that these efforts will provide opportunities to include new sources of in situ SWE information such as data collected at long-term experimental sites maintained by academic partners.

### Appendix A: Previous use of the 2019 CHSSD update

The 2019 CHSSD update has been produced by Brown et al. (2019). This dataset has been used by different research groups in support of model evaluation, climate monitoring and development of innovative algorithms. A search was carried out on Google Scholar (last access: 20 July 2020) to list all the studies that refer to the paper by Brown et al (2019). Each study was then considered and all the studies that used the 2019 CHSSD update were listed in Table A1.

**Table A1: List of the studies that cited and used the 2019 CHSSD update**

| Reference | Use of the 2019 CHSSD Update |
|---|---|
| Gasset et al (2021) | Evaluation of snow simulations (SWE, SD, density) in a reanalysis product |
| Luojus et al (2021) | Evaluation and bias-correction of a satellite-based SWE product over the Northern Hemisphere |
| Mortimer et al. (2020) | Evaluation of long term-gridded snow products over the Northern Hemisphere |
| Ntokas et al. (2021) | Estimation of SWE from SD using artificial neural networks |
| Pulliainen et al. (2020) | Evaluation of long term-gridded snow products over the Northern Hemisphere |
| Royer et al. (2021a) | Development of a new northern snowpack classification in Canada |
| Royer et al. (2021b) | Evaluation of snow simulations (SD, density) in the Arctic |
| Venäläinen et al (2021) | Development of snow density field to improve gridded SWE products over the Northern Hemisphere |

## Author contribution

VV, CM and RB initiated the 2021 update of the CHSSD leading to CanSWE. VV coordinated the update effort. VV and CM reached out partners agencies to obtain snow data and processed them. MB developed the routines for the automatic detection of duplicates and conducted systematic identification of duplicates. CM developed the quality control routines and data flag consolidation. LA identified duplicates in the 2019 update of the CHSSD and systematically tested the intermediate versions of CanSWE and identified remaining issues that were then corrected. All authors contributed to the preparation of the manuscript. We thank an anonymous reviewer and Charles Fierz for their careful reading and useful comments, which improved the manuscript.

## Competing interest

The authors declare that they have no conflict of interest.

## Acknowledgements

The following agencies are gratefully acknowledged for the high quality of their snow data collection programs and for providing historical data to the new CanSWE dataset: Alberta Environment and Parks, British Columbia Ministry of Environment and partners, Environment New Brunswick, Government of Newfoundland and Labrador, Government of Northwest Territories, Manitoba Hydro, Meteorological Service of Canada (ECCC), Ontario Ministry of Natural Resources and Forestry and partners, Ontario Power Generation, Saskatchewan Water Security Agency and Yukon Water Resource Branch. Many thanks are expressed to the field observers collecting manual snow survey data across Canada and the persons in charge of the maintenance of automatic stations deployed across the country. Hydro Québec's https://www.hydroquebec.com, last access: 19 April 2021) data are available under the terms of a Creative Commons Attributions – Non Commercial – Share A Like 4.0 International License (https://creativecommons.org/licenses/by-nc-sa/4.0/, last access: 19 April 2021).

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
