# Peer review of "Canadian historical Snow Water Equivalent dataset (CanSWE, 1928-2020)"

_Earth System Science Data, 2021_

## Referee Comment (RC2)

470

475

[referee-annotated manuscript omitted]

---

## Author Comment (AC1)

**Answer to Reviewer 1 ESSD-2021-160**

We thank Reviewer 1 for their comments. We provide here our responses to those comments and describe how we addressed them in the revised manuscript. The original reviewer comments are in normal black font while our answers appear in blue font.

**General comments**

This manuscript describes a pan-Canadian data set on snowpack water equivalent (SWE), along with snow depth and snow density for observations for which snow depth has been reported in addition to SWE. The data set is an updated version of the Canadian Historical Snow Survey Data (CHSSD) archive, which has been used in a number of research studies since its publication in 2019. The current version corrects a number of issues in the earlier version, incorporates additional data sets, and applies a consistent quality control protocol. The steps involved in the updating are all clearly described and logical.

Based on the usage of the earlier version by the international community, I anticipate that this updated version will be an important resource for a range of studies related to atmospheric and climate science, cryospheric science, hydrology and ecology. I have a few suggestions for some additional information and technical corrections, as outlined below.

**Specific comments**

The introduction seems to me a bit long, and I wonder if all of the information is necessary in the context of introducing the data set. I would suggest that the authors consider ways to shorten it. For example, perhaps some of the information on measurement approaches in the first two paragraphs could be summarized in a table. That table could also be referred to later in the manuscript in relation to the metadata.

We thank Reviewer 1 for this comment. During the revision process, we consider the possibility of shortening the introduction. However, based on several comments from Charles Fierz (Reviewer 2) asking to add information regarding manual SWE measurements, we decided to keep in the introduction the two sections that describes the measurement approaches and the measurement networks used by different countries. We believe these two sections provide useful general information for the readers interested in snow dataset.

To help set the motivation for producing the current data set, it may be useful to add a couple of sentences to the introduction about the use of the earlier CHSSD by the international community. For example, a Web of Science search on the article by Brown et al. (2019, Atmos. Ocean) showed that it has already been cited eight times.

The typical different uses of SWE datasets were already described in the introduction of the initial paper (L46-L59). However, as pointed out by Reviewer 1, this description was not specific to CanSWE. As suggested, we added this information in the revised paper. The papers that cited and used the 2019 CHSSD update by Brown et al (2019) were identified using a search on Google Scholar (last access 20 July 2021) and listed in the table below. To limit the length of the introduction, this table has been added to Appendix A of the revised manuscript and the text in the introduction refers to this table.

| Reference | Use of the 2019 CHSSD Update |
|---|---|
| Gasset et al. (2021) | Evaluation of snow simulations (SWE, SD, density) in a reanalysis product |
| Luojus et al. (2021) | Evaluation and bias-correction of a satellite-based SWE product over the Northern Hemisphere |
| Mortimer et al. (2020) | Evaluation of long term-gridded snow products over the Northern Hemisphere |
| Ntokas et al. (2021) | Estimation of SWE from SD using artificial neural networks |
| Pulliainen et al. (2020) | Evaluation of long term-gridded snow products over the Northern Hemisphere |
| Royer et al. (2021a) | Development of a new northern snowpack classification in Canada |
| Royer et al. (2021b) | Evaluation of snow simulations (SD, density) in the Arctic |
| Venäläinen et al. (2021) | Development of snow density field to improve gridded SWE products over the Northern Hemisphere |

Text added to the introduction: ***The 2019 CHSSD update has been used in numerous studies (see Table A1 for a complete list). However,*** *researchers working with the 2019 CHSSD update …..*

Figures 4 to 7 provide a good overview of the spatial and temporal coverage that will be useful for potential users. The only suggestion I would have for additional figures would be one showing the elevational distribution of observations in relation to hypsometry, perhaps at a provincial or regional scale (e.g., based on the national level ecoregions; see https://open.canada.ca/data/en/dataset/ade80d26-61f5-439e-8966-73b352811fe6). As a researcher who focuses on the mountainous regions of western Canada, I believe that it is important for users of SWE data to appreciate that most of our observations represent mid-elevation locations below treeline.

Thanks for this suggestion. A new figure has been added to the revised manuscript (see below and Fig. 6 in the revised manuscript). For each province and territory, it compares the distribution of the elevation of the stations with the hypsometry of the province/territory. The hypsometry has been derived from the Global Multi-resolution Terrain Elevation Data 2010 (GMTED2010, https://www.usgs.gov/core-science-systems/eros/coastal-changes-and-impacts/gmted2010) at 30 arc-seconds reprojected to the Canada Albers Equal Area Conic projection at 250-m grid spacing.

[Figure]

A paragraph describing this figure has been added to the text:

*Figure 6 compares the distribution of the station elevation with the hypsometry in each province and territory. The hypsometry has been derived from the Global Multi-resolution Terrain Elevation Data 2010 (https://www.usgs.gov/core-science-systems/eros/coastal-changes-and-impacts/gmted2010, last access 21 July 2020) at 30 arc-seconds reprojected to the Canada Albers Equal Area Conic projection at 250-m grid spacing. Figure 6 shows that the elevation coverage provided by the stations varies greatly from one region to another. A representative coverage is found in provinces of Eastern Canada (Quebec, New Brunswick, Nova Scotia). On the other hand, in British Columbia and Alberta, SWE measurement sites tend to be located at higher elevation than the average terrain to provide relevant information on snow cover in mountainous headwater catchments. Large differences between the station elevation coverage and the hypsometry are also found in Nunavut and Saskatchewan. They are associated with sparse spatial coverage in the elevated inland parts of Nunavut and in the low-elevation northern part of the province in Saskatchewan.*

It would be useful for potential users to have information about the different types of snow samplers used in the different regions of Canada, If such information is available. For example, Goodison et al (1987, https://doi.org/10.4296/cwrj1202027) reported that many samplers used in Canada overmeasure by varying amounts.

The different agencies that provided snow data to CanSWE have been contacted to get information about the types of snow samplers that they are using. The information has been gathered in the table below. This table has been included in revised manuscript (Table 2 in the new version)

**Table 2: Equipements for manual (snow samplers) and automatic SWE measurements used by each agency that provided snow measurements for CanSWE.**

| Agency | Manuel stations Snow samplers | Automatic stations |
|---|---|---|
| Yukon Water Resources Branch | Federal sampler | - |
| Government of Northwest Territories | ESC-30 sampler | - |
| Meteorological Service of Canada (ECCC) | ESC-30 sampler | - |
| British Columbia Ministry of Environment | Federal sampler | Snow pillows |
| Alberta Environment and Parks | Federal sampler | Snow pillows |
| Saskatchewan Water Security Agency | Prairie sampler | - |
| Manitoba Hydro | Federal sampler | - |
| Ontario Power Generation | Federal sampler (at most sites), ESC-30 sampler at some sites | - |
| Ontario Ministry of Natural Resources and Forestry | | - |
| Hydro Québec | Federal sampler | Passive gamma radiation sensors |
| Government of New Brunswick | Federal sampler | - |
| Government of Newfoundland and Labrador | Federal sampler | Passive gamma radiation sensors |

The following text has been added to Section 2.1 and refers to papers discussing the impact of snow samplers on the accuracy of SWE measurements:

*Most of the agencies use the Federal snow sampler whereas the Prairie and the ESC-30 samplers are used in regions of shallow snowpack such as the Prairies or the Arctic (Table 2). The Federal snow sampler is a small-diameter and multi-section sampler design to aid sampling in deep snowpack whereas the Prairie and the ESC-30 samplers present large diameters tubes to maximize snow collection in shallow snow cover and increase measurement accuracy (Dixon and Boon, 2012). More details about the impact of sampler type on uncertainties in SWE measurements are given in Godison et al. (1987) and Lopez Moreno et al. (2020).*

Another methodological point that may be useful to mention, if information is available, relates to the siting of snow courses and snow pillows. For example, some snow pillows I have seen in British Columbia are located in small forest openings, such that I suspect that they tend to accumulate snow more like an open site and melt at rates more like forested sites.

We fully agree with Reviewer 1 that information about the location of snow measurement sites is crucial to better understand the snowpack dynamics at each site. In particular, this is critical when evaluating the output of a distributed snowpack model. However, such information is currently not available directly from the different agencies across Canada. Adding such information would require an extensive work in close collaboration with the agencies. It was not possible to complete this work during the time allocated for the review of this paper. Another solution would have been to extract the information about the vegetation cover from a high-resolution vegetation database such as the product from Hansen et al. (2013). However, uncertainty with the accuracy and precision of the station coordinates may affect this extraction (see our answer to the next comment). Therefore, at this stage, information about the sitting of snow measurement sites in CanSWE is not available. We will work to add this information into future versions of CanSWE.

The absence of information about the sitting of the snow measurements sites in CanSWE is now explicitly mentioned in the revised manuscript (Section 2.2):

*Information about the sitting of the snow measurement sites (e.g., open terrain, below forest, clearing) is not available in the present version of CanSWE and will be added to future version of the dataset.*

Hansen, M. C., Potapov, P. V., Moore, R., Hancher, M., Turubanova, S. A. A., Tyukavina, A., and Kommareddy, A.: High-resolution global maps of 21st-century forest cover change, Science, 342, 850–853, https://doi.org/10.1126/science.1244693, 2013.

When comparing model output to SWE observations, it is important for users to understand the accuracy of the locational coordinates in order to extract simulated SWE from a model unit that is representative of the monitoring location. If possible, I suggest that the authors add some information about the typical horizontal and vertical accuracies of the coordinates.

Thank you for this comment. We agree that accurate coordinates and elevations underpin the usefulness of any in situ dataset, including CanSWE. Coordinates are provided by contributing agencies and may have been obtained from a variety of sources including topographic maps, handheld GPS or a differential GPS systems the accuracy of each method varies but this information is not provided by the contributing agencies. As such, we are only able to speak to coordinate precision and the accuracy of the USGS NED which is used when elevation is not provided by an agency.

That said, in terms of precision, coordinates are most often reported in decimal degrees with two to seven decimal places depending on the agency. However, the level of precision can be misleading if the original coordinates were reported in degrees, minutes, seconds. Elevations are reported to the nearest metre or tens of metres, depending on the agency.

Many agencies did not include elevation in their metadata. In these instances, elevations were obtained from the United States Geological Survey's National Elevation 190 Dataset (USGS NED) (Gesch et al. 2002). Vertical accuracy of the 1 arc second 2013 release over Canada is 3.53 m (Gesch et al. 2014). This Root Mean Squared Error value was obtained via comparisons with 578 reference control points. The vertical

accuracy also varies by location and source data and the set of reference control points did not include any locations north of ~50°N except for a few rivers in Quebec and Labrador and some costal locations in Atlantic Canada. We took the elevation of the grid cell that intersected with each point, regardless of where the point fell within the grid cell. Topographic variability within a grid cell (below 1 arc second) would add additional uncertainty to the reported elevation. Despite these caveats, we have included the reported accuracy of 3.53 m in our revised manuscript.

Reference:

Gesch, D.B., Oimoen, M.J., and Evans, G.A., 2014, Accuracy assessment of the U.S. Geological Survey National Elevation Dataset, and comparison with other large-area elevation datasets—SRTM and ASTER: U.S. Geological Survey Open-File Report 2014–1008, 10 p., http://dx.doi.org/10.3133/ofr20141008.

Suggested revision:

*When elevation was not present in the metadata from the originating agency it was extracted from the United States Geological Survey's National Elevation Dataset (USGS NED, Gesch et al., 2002) at the position corresponding to the location of the snow survey site. The USGS's NED covers all Northern America at 30-m resolution (except parts of Alaska)* **and has a vertical accuracy of 3.53 m over Canada (Gesch et al., 2014).**

**Technical corrections**

line 104. I suggest reorganizing the sentence to avoid beginning with "91." The revised sentence is written as follows: *A total of 91*  *stations were identified and were manually checked*

line 192. Should "Northern America" be "North America" – i.e., Mexico, USA and Canada? Indeed, it was a mistake in the original version of the paper. It has been corrected.

line 192. Change "expect" to "except." Correction included

line 207. Change "was" to "were" to be consistent with earlier usage of "data" as plural. Correction included

line 283. Insert "of" between "majority" and "the." Correction included

---

## Author Comment (AC2)

**Answer to Charles Fierz ESSD-2021-160**

We thank Charles Fierz for his very detailed review. We provide here our responses to his comments and describe how we addressed them in the revised manuscript. His original comments are in normal black font while our answers appear in blue font.

**General comments**

This paper presents an update of the 2019 Canadian Historical Snow Survey dataset (CHSSD). Errors in metadata as well as a large amount of duplicate data led to this overhaul. Furthermore, the dataset was renamed 'Canadian historical SWE dataset (CanSWE)' to reflect the inclusion of automated data and the weight put on water equivalent of snow cover (SWE) data. The paper is well written and well structured, describing clearly the processing steps as well as the cleaning procedures used. Great work!

The paper is timely as it emphasizes the need for standardized formats and procedures, "Python routines specific to each agency and corresponding data format were written to process the data and metadata and arrange them in a consistent NetCDF format". While I very much welcome this effort towards standardization I may have wished an even larger convergence towards WMO standards, for example regarding terminology (see comments below). However, as the authors conclude that "Regular updates are required to make such datasets useful for the community.", I would strongly suggest following WMO standards even more closely in future. For example, I anticipate a standardized WMO NetCDF format will emerge soon to which the CanSWE format may comply too.

In summary I very much appreciate the effort put in homogenising the Canadian snow survey data sets and therefore recommend accepting the paper after the authors addressed the issues below and do some minor revisions as suggested in the annotated manuscript.

The paper has been revised according to the suggestions made below by C. Fierz. In particular, we improved the terminology and the list of references to better follow the WMO standards. This also led to a new version of the CanSWE netcdf file and its related documentation in English and French on the Zenodo data portal (see Canadian historical Snow Water Equivalent dataset (CanSWE, 1928-2020)] Zenodo - Version-v2).

**Specific comments**

• General introduction about snow courses and snow surveys, lines 23-60:

I miss a clear statement whether you only consider snow courses as relevant measurements of water equivalent of snow cover. While on lines 39-40 you state that, "SWE observation networks using different measurements methods have been deployed at a national scale in various countries to provide valuable in situ information.", the following lines only include countries performing snow surveys, which are most typical in North America indeed, but not everywhere. I would suggest including a short paragraph on other methods, referring as you do to the 'European Snow Booklet' but also to WMO-No.8 (WMO, 2018) and also to LópezâMoreno et al. (2020).

We agree with C. Fierz that the introduction of the initial manuscript was not clear about the manual techniques used to measure SWE at single point locations. We do not consider that snow courses are the

only relevant manual measurements of SWE. Therefore, the introduction has been slightly adapted to make it clear:

Manual SWE measurements typically consist of single point measurement (snow pit or single measurement carried out with a snow tube) or multi-point gravimetric snow surveys (also known as snow transects or snow courses) collected along a pre-determined transect (WMO, 20108, Lopez Moreno et al., 2020).

National SWE measurements relying on manual methodssnow survey are also available in several European countries: such as Finland, Estonia, Ukraine and Turkey use for example snow courses whereas countries such as Germany or Czech Republic rely on single point measurement-(Haberkorn, 2019).

We also added the reference to Lopez Moreno et al (2020) when mentioning the uncertainty in SWE measurement associated with the different type of snow samplers (see one comment of Reviewer 1):

Most of the agencies use the Federal snow sampler whereas the Prairie and the ESC-30 samplers are used in regions of shallow snowpack such as the Prairies or the Arctic (Table 2) ... More details about the impact of sampler type on uncertainties in SWE measurements are given in Godison et al. (1987) and Lopez Moreno et al. (2020).

• Section 3, Quality control of the final dataset :

The homogenization of data quality flags shows how important standardized report practices are.

Do I understand correctly that a H or W flag automatically results in a D flag? In other words, a quality flag set at one step influences the final number of flagged values?

In an effort to reduce the number of variables in the dataset, there are only QC flags for snw and snd. SWE and SD ranges are applied first and then the final bulk snow density calculated. SWE and SD values failing their respective tests are set to *nan* and no density value is calculated. This means that if a SWE or SD value fails its range test, no density value is calculated (set to *nan*).

| Condition                | snw           | snd           | den | Qc_flag_snw | Qc_flag_snd |
|--------------------------|---------------|---------------|-----|-------------|-------------|
| SWE > range              | nan           | Numeric value | nan | W           |             |
| SD > range               | Numeric value | nan           | nan |             | Н           |
| SWE and SD outside range | nan           | nan           | nan | W           | Н           |
| Density outside range    | nan           | nan           | nan | D           | D           |

 Table R2. Application of range threshold flags in order they are applied during QC for various conditions.

We made the following clarifications to the text:

When a record failed the SWE (SD) threshold but not the SD (SWE) threshold only the SWE (SD) value was set to NaN; the corresponding density value was also set to NaN and a W or H flag assigned to these records (Table 5). When a record failed the bulk snow density threshold SWE, SD and bulk snow density were set to NaN; a D flag was assigned to these records (Table 5).

• Section 4, Data availability and Table 6, lines 328-336:

From my perspective, except snw, snd, den, most of the entries in this table are observational metadata in the sense defined by WMO (WMO-No. 49, Volume I and WMO-No. 1160). It may be worth considering whether the WMO standard could be implemented in a future version of CanSWE, both regarding the name of metadata and data as well as within the NetCDF file. A note on this in the text would be appreciated.

As suggested by Charles Fierz, the term "observational metadata" is used in Table 6 and horizontal separators has been added in Table 6 to help distinguishing between the types of variables. The following sentence has also been added in Section 4:

*Future versions of CanSWE will include updated names for the observational metadata to follow the WMO standards (WMO, 2019b)*

**Comments on terminology**

• 'snow water equivalent' vs "water equivalent of snow cover", lines 1 (title), 10, 23, 336 (table 6), 338:

Consider switching to new terminology introduced in WIGOS Metadata Standard (WMO-No. 1160) and WMO-No. 8, keeping SWE as abbreviation.

As recommended by C. Fierz, the new terminology is used in the revised paper. In particular, we insisted on the use of the new terminology in the first sentence of the introduction of the revised manuscript:

*Reliable in situ information of snow water equivalent (SWE) or more precisely water equivalent of snow cover according to WMO (2018)(SWE) – t*

We decided to keep the title of the paper unchanged to be consistent with the name of the dataset that has been already published on Zenodo. On Zenodo, the variable names have been updated in the Netcdf file to use the new terminology. The readme files in English and French have also been changed.

• 'the depth of water that would be produced if all the snow melted', line 10:

Both the WIGOS Metadata Standard & WMO-No. 8, Vol 2, p. 13 state: "Water equivalent of snow cover (SWE) is the vertical depth of water that would be obtained if the snow cover melted completely, which equates to the snow-cover mass per unit area."

The reference definition for SWE from WMO-No. 8 is now used in the abstract and at the beginning of the introduction:

Abstract: In situ measurements of snow water equivalent (SWE) – the vertical depth of water that would be obtained if all the snow cover melted completely – ...

Introduction: Reliable in situ information of snow water equivalent (SWE) – the vertical depth of water that would be obtained if the snow cover melted completely, which equates to the snow-cover mass per unit area (WMO, 2018) – ...

| • | '[derived] | bulk | snow | density': |
|---|------------|------|------|-----------|
|---|------------|------|------|-----------|

I very much welcome you using 'bulk snow density'!

- Consider defining it on line 211 as ratio of SWE to SD – and also in the abstract, line 14 – Consider adding 'derived' only in tables (4, 5, and 6) but not in the text (line 339)

- Use it throughout the text (see lines 245, 246, 250, 278, and 344) and not on line 255 as it is referring to a range.

Bulk snow density is new defined in the abstract: *Snow depth (SD) and bulk snow density (defined as the ratio of SWE to SD) are also included when available.*

It is also defined in the main text at the end of Sect. 2.2: *Finally, where both SWE and SD measurements were available, bulk snow density was calculated from the ratio of SWE to SD and included in the final database.*

We followed the recommendation of C. Fierz regarding the use of the term "derived" and we only kept it the different tables. It is not use anymore in the text. The term "Bulk snow density" is used systematically in the revised manuscript (except when it is referring to a range as suggested by C. Fierz).

**Comments from the pdf**

P1 L 23-24: Even though the citation is correct :-), I would now prefer you referring to WMO-No. 8 and the definition therein, as detailed in my 'Comments on terminology'.

The reference to WMO (2018) is used in the manuscript.

P 1 L29: Here I would refer to WMO-No. 8 too As suggested in a comment made below about the reference section, the reference to WMO (2008) has been replaced by a reference to WMO (2018).

P2 L 33: Consider adding ref by Craig Smith?

Smith, C. D., Kontu, A., Laffin, R., and Pomeroy, J. W.: An assessment of two automated snow water equivalent instruments during the WMO Solid Precipitation Intercomparison Experiment, The Cryosphere, 11, 101–116, https://doi.org/10.5194/tc-11-101-2017, 2017.

Reference included.

**P2 L 35: Kodama Correction included**

P 2: 37-38 I would say that GNSS includes GPS. Thus, is there a need to cite twice the same authors? On the other side, the paper below by L. Steiner gives a view on low-cost GNSS sensors.

Steiner, L., Meindl, M., Fierz, C., Marty, C., and Geiger, A.: Monitoring snow water equivalent using lowcost GPS antennas buried underneath a snowpack, in: 2019 13th European Conference on Antennas and Propagation (EuCAP), 2019 13th European Conference on Antennas and Propagation (EuCAP), 31 March-5 April 2019, Krakow, Poland, 1–5, 2019. Thanks for this comment. We agree that GNSS includes GPS. Therefore, the sentence has been rephrased. We kept only the reference to Henkel et al. (2018) using GNSS sensors and added the reference suggested by C. Fierz:

Finally, SWE can be automatically derived by analysis of the signal from Global Navigation Satellite System receivers (Henkel et al., 2018, Steiner et al., 2019). or Global Positioning System receivers (Koch et al., 2019).

P 2 L49: Reference missing! I guess you meant:

Brun, E., Vionnet, V., Boone, A., Decharme, B., Peings, Y., Valette, R., Karbou, F., and Morin, S.: Simulation of Northern Eurasian Local Snow Depth, Mass, and Density Using a Detailed Snowpack Model and Meteorological Reanalyses, J. Hydrometeor., 14, 203–219, https://doi.org/10.1175/JHM-D-12-012.1, 2013.

**Thanks a lot! Reference added.**

**P 3 L73: Is there public access to it?**

The data were distributed on CD-ROMs (see https://openpolar.no/Record/ftdatacite:oai:oai.datacite.org:9832197) and the integrality of the dataset has then been transferred into the dataset published by Brown et al. (2019).

**P4 L 111: How many duplicate dates are needed to call a station a duplicate? Does this suffice to identify a station as such?**

A minimum number of 10 dates was considered. As explained at L120-122 of the original manuscript, additional information on station locations and names was also considered to decide if two stations should be matched. This selection was made manually. We added to the text the information about the minimal number of dates.

P5 L 121 : See comment above, line 111. See our previous answer.

P5 L 123: I assume a matching station was not further considered as either a reference station or a neighbor to another station, correct?

Exactly. This is an iterative process so that if a merge key has already been assigned to a given station, this station is not considered in the neighborhood analysis.

P5 L 124: I think I undestand what 'reference' and 'matching' stations mean. However, the reader needs to be attentive and definitions might be helpful.

The term 'reference station' has been replaced by the term 'inspected station' and this term is now defined clearly in the text:

For each station in the CHSSD (referred here as 'inspected station'), all stations within a 5-km radius were identified. Each group of neighbouring stations was then manually inspected for similarities in (i) snow measurements for matching dates (at least 10), (ii) station location and (iii) station name. In most cases, all three of the criteria were satisfied to trigger a decision on whether a duplicate was identified. When a duplicate was identified, the inspected station and its matching neighbors were assigned a unique merging key to be used in subsequent consolidation. If no similar stations to thea inspected reference

station were identified in a group of neighbouring stations, the inspected reference station was assigned its own merging key to aid in future updates to the CHSSD.

P6: Figure 2: Why not call it 'cleaned 2019 CHSSD'? This would better match the denomination in Figure 1.

Thank you for the suggestion. Figure has been updated.

P6 L 158: If possible, it would be interesting to know what the problem is such that standardization efforts elsewhere can avoid those.

The extensive historical snow dataset provided by the province of Manitoba consisted of scanned field books in pdf format over the period (1952-2017) and of a mix a scanned field books and Excel spreadsheets from 2018. The inclusion of this dataset into CanSWE would have required a significant effort that was beyond the time that was allocated to the 2021 update of CanSWE. In addition, such work needs to be done in close collaboration with the agency providing the data to benefit from its guidance. We hope this extremely valuable dataset will be processed in the future by the province of Manitoba to allow its ingestion into CanSWE.

**P7 L 178: ranging instruments Correction included**

P8 Table 2: To my knowledge this table is currently only implemented in BUFR since about 2018! You need to adapt the reference to the Manual on Codes, as indicated in the references.

As recommended, the citation WMO (2019) has been adapted in the revised manuscript to correspond to WMO Manual on Codes.

P 10 Table 3: There are 12 flags in the table. On line 228 above you state the final dataset contains 25 flags. Please comment and/or rectify.

Correction made. There are 10 agency flags for SWE and 8 for SD, not 14 and 11 as stated on line 228, now that the L and Q flags have been removed (see below).

P10 Table 3: Quite fuzzy definition. Can you expand please, for example "early morning or late afternoon"

Definition clarified. The term 'Manual snow survey conducted outside the nominal sampling period.' is now used. Early or late sampling indicates that a manual survey was made outside of the nominal sampling period; on the order of days or weeks.

P10 Table 3: Not sure I understand this correctly? Would have been meaningful to make it the 'M' flag.

The L and Q flags were relics of the older CHSSD and have been removed from the table and from the netcdf file (see the new file on Zenodo).

P10 Table 4: Add the reference to Leys et al. (2018) as a note here. Reference added as a footnote.

P10 Table 4: See comment on line 245. See our answer below.

P11 L 244: Please stick to kg m-2 Correction included.

P 11 L 245: I find it a little bit bold to call these ranges 'common'. That may be true for Canada, but not in general. Is the range 50 - 700 kg m-3 stringent at all? Both thresholds look quite 'extreme' to me. Could you please expand on this? For example, how many bulk densities in the dataset pertain to the range 50 to 200 kg m-3 and how many to the range 500 to 700 kg m-3? Is the lower range linked to low SD-values? Etc.

We agree with Charles Fierz that the terms "common" and "stringeant" used in the initial manuscript were not totally appropriate. The figure below shows the distribution of bulk snow density obtained at different sites before applying the range thresholding on density. The median bull snow density is 254 kg m-3 for manual snow survey, 249 kg m-3 at the location of the GMON sensors and 325 kg m-3 at the location of the snow pillows. Among the manual snow survey, 2.2 % of the bulk snow density are in the range 500–700 kg m-3 and 26.3 % in the range 50–200 kg m-3.

Figure: Distribution of bulk snow density before quality control.

The aims of the range thresholding for bulk density used in this paper is to identify SWE-SD pairs that are likely erroneous and to remove the corresponding values from the final dataset. Given the large variety of snow climates covered by CanSWE, we used an extended range of potential values for bulk snow density (25-700 kg m-3). A user of the dataset may want to use a more advanced QC with a density range that depends on the location and the time of year.

The revised manuscript has been adjusted as follows:

Range thresholds were used to identify spurious records in both automated and manual measurements. We adopted the thresholds outlined in Brown et al. (2019) for SWE and SD ( $0 - 3000 \text{ kg m}^{-2} \text{mm}$ ,  $0 - 8000 \text{ kg}}{\text{m}^{-2} \text{-mm}}$  for mountain) but a slightly more restrictivestringent range of 25 - 700 kg m-3 (as opposed to  $50 - 1000 \text{ kg m}^{-3}$ ) for bulk snow density. These ranges are based on common ranges for SWE and, SD <del>and density</del> from the literature (see Braaten, 1998). The range thresholding applied to bulk snow density aims at identifying SWE-SD pairs that are likely erroneous.

**P 11 L 246-247: What does this mean in terms of elevation? I miss in general an altitudinal distribution of the stations.**

A figure detailing the altitudinal distribution of the stations has been added to the revised version of the paper (see below). The definition of mountains used for CanSWE is extremely simple and has been selected to be consistent with the earlier versions of the CHSSD. For CanSWE, it means that specific range thresholds for SWE and SD are used for all the stations located in British Columbia and Yukon, for the stations located in the part of the Canadian Rockies in Alberta and for some stations located in the Northwestern Territories. A more relevant distinction between mountainous and non-mountainous regions (e.g., Karagulle et al., 2017) could be used for QC in future versions of CanSWE. A sentence has been added in Sect. 2.3 to mention this possibility:

**This definition is very simple and more advanced definitions (e.g., Karagulle et al., 2017) may be considered in future version of CanSWE.**

Reference

Karagulle, D., Frye, C., Sayre, R., Breyer, S., Aniello, P., Vaughan, R., & Wright, D. (2017). Modeling global Hammond landform regions from 250-m elevation data. *Transactions in GIS*, *21*(5), 1040-1060.

P 11 L 248: Table 4? Modification included

P11 L 250: bulk density Modifications included

P 11 Figure 3: I already highly appreciate you using kg m-2. Could '-2' be written as superscript though?

Figure 3 has been updated.

---

## Author Response (AR1)

Vincent Vionnet Environment and Climate Change Canada, Dorval, Canada Tel.: +1 438 366 0148 Email: vincent.vionnet@ec.gc.ca

August 20th, 2021

Dear ESSD Editor,

Please find enclosed a revised version of the manuscript ESSD-2021-160. As requested by the two reviewers, we made changes to the paper to improve its quality. In particular, we added a new figure that describes the altitudinal distribution of the snow measurement sites within each province and territory in Canada. We also created two new tables to describe (i) the type of snow samplers used by the different agencies across Canada and (ii) the different studies that used 2019 version of the Canadian Historical Snow Survey Dataset that serve as reference for the new CanSWE dataset. The terminology has also been adapted to be consistent with the terminology recommended by the World Meteorological Organization. This led to the generation of a new version of the CanSWE Netcdf file and the associated documentation. This version is already available to the community and identified with a unique DOI issued by the Zenodo data management platform (see https://zenodo.org/record/5217044).

Our detailed answers to the reviewers are described in the following pages of this document.

Thank you in advance for taking this new version of our paper into consideration,

Sincerely yours,

Vincent Vionnet and co-authors.

**Answer to Reviewer 1 ESSD-2021-160**

We thank Reviewer 1 for their comments. We provide here our responses to those comments and describe how we addressed them in the revised manuscript. The original reviewer comments are in normal black font while our answers appear in blue font.

**General comments**

This manuscript describes a pan-Canadian data set on snowpack water equivalent (SWE), along with snow depth and snow density for observations for which snow depth has been reported in addition to SWE. The data set is an updated version of the Canadian Historical Snow Survey Data (CHSSD) archive, which has been used in a number of research studies since its publication in 2019. The current version corrects a number of issues in the earlier version, incorporates additional data sets, and applies a consistent quality control protocol. The steps involved in the updating are all clearly described and logical.

Based on the usage of the earlier version by the international community, I anticipate that this updated version will be an important resource for a range of studies related to atmospheric and climate science, cryospheric science, hydrology and ecology. I have a few suggestions for some additional information and technical corrections, as outlined below.

**Specific comments**

The introduction seems to me a bit long, and I wonder if all of the information is necessary in the context of introducing the data set. I would suggest that the authors consider ways to shorten it. For example, perhaps some of the information on measurement approaches in the first two paragraphs could be summarized in a table. That table could also be referred to later in the manuscript in relation to the metadata.

We thank Reviewer 1 for this comment. During the revision process, we consider the possibility of shortening the introduction. However, based on several comments from Charles Fierz (Reviewer 2) asking to add information regarding manual SWE measurements, we decided to keep in the introduction the two sections that describes the measurement approaches and the measurement networks used by different countries. We believe these two sections provide useful general information for the readers interested in snow dataset.

To help set the motivation for producing the current data set, it may be useful to add a couple of sentences to the introduction about the use of the earlier CHSSD by the international community. For example, a Web of Science search on the article by Brown et al. (2019, Atmos. Ocean) showed that it has already been cited eight times.

The typical different uses of SWE datasets were already described in the introduction of the initial paper (L46-L59). However, as pointed out by Reviewer 1, this description was not specific to CanSWE. As suggested, we added this information in the revised paper. The papers that cited and used the 2019 CHSSD update by Brown et al (2019) were identified using a search on Google Scholar (last access 20 July 2021) and listed in the table below. To limit the length of the introduction, this table has been added to Appendix A of the revised manuscript and the text in the introduction refers to this table.

| Reference            | Use of the 2019 CHSSD Update                                         |  |  |  |
|----------------------|----------------------------------------------------------------------|--|--|--|
| Gasset et al. (2021) | Evaluation of snow simulations (SWE, SD, density) in a reanalysis    |  |  |  |
|                      | product                                                              |  |  |  |
| Luojus et al. (2021) | Evaluation and bias-correction of a satellite-based SWE product over |  |  |  |
|                      | the Northern Hemisphere                                              |  |  |  |
| Mortimer et al.      | Evaluation of long term-gridded snow products over the Northern      |  |  |  |
| (2020)               | Hemisphere                                                           |  |  |  |
| Ntokas et al. (2021) | Estimation of SWE from SD using artificial neural networks           |  |  |  |
| Pulliainen et al.    | Evaluation of long term-gridded snow products over the Northern      |  |  |  |
| (2020)               | Hemisphere                                                           |  |  |  |
| Royer et al. (2021a) | Development of a new northern snowpack classification in Canada      |  |  |  |
| Royer et al. (2021b) | Evaluation of snow simulations (SD, density) in the Arctic           |  |  |  |
| Venäläinen et al.    | Development of snow density field to improve gridded SWE products    |  |  |  |
| (2021)               | over the Northern Hemisphere                                         |  |  |  |

**Text added to the introduction: *The 2019 CHSSD update has been used in numerous studies (see Table A1 for a complete list). However, researchers working with the 2019 CHSSD update .....**

Figures 4 to 7 provide a good overview of the spatial and temporal coverage that will be useful for potential users. The only suggestion I would have for additional figures would be one showing the elevational distribution of observations in relation to hypsometry, perhaps at a provincial or regional scale (e.g., based on the national level ecoregions; see https://open.canada.ca/data/en/dataset/ade80d26-61f5-439e-8966-73b352811fe6). As a researcher who focuses on the mountainous regions of western Canada, I believe that it is important for users of SWE data to appreciate that most of our observations represent mid-elevation locations below treeline.

Thanks for this suggestion. A new figure has been added to the revised manuscript (see below and Fig. 6 in the revised manuscript). For each province and territory, it compares the distribution of the elevation of the stations with the hypsometry of the province/territory. The hypsometry has been derived from the Global Multi-resolution Terrain Elevation Data 2010 (GMTED2010, https://www.usgs.gov/core-science-systems/eros/coastal-changes-and-impacts/gmted2010) at 30 arc-seconds reprojected to the Canada Albers Equal Area Conic projection at 250-m grid spacing.

---

## Author Response (AR2)

Vincent Vionnet
Environment and Climate Change Canada, Dorval, Canada
Tel.: +1 438 366 0148
Email: vincent.vionnet@ec.gc.ca

September 2nd, 2021

Dear ESSD Editor,

Please find enclosed the accepted version of the manuscript ESSD-2021-160. As requested in your comment, the corrected version of Figure 3 is now used in the document and has been provided in the zip archive containing all the figures of the paper.

Thank you handling the review process of this paper.

Sincerely yours,

Vincent Vionnet and co-authors.